# A Few-Shot Learning-Based Reward Estimation for Mapless Navigation of Mobile Robots Using a Siamese Convolutional Neural Network

Vernon Kok [1], Micheal Olusanya [2,*] and Absalom Ezugwu [1]

1 School of Mathematics, Statistics and Computer Science, University of Kwa-Zulu Natal, Durban 4041, South Africa; 221116607@ukzn.ac.za (V.K.); Ezugwua@ukzn.ac.za (A.E.)
2 School of Natural and Applied Sciences, Sol Plaatje University, Kimberley 8300, South Africa
* Correspondence: michael.olusanya@spu.ac.za

**Abstract:** Deep reinforcement learning-based approaches to mapless navigation have relied on the distance to the goal state being known a priori or that the distance to the goal can be obtained at each timestep. In artificial or simulated environments, obtaining the distance to the goal is considered a trivial task. Still, when applied to a real-world scenario, the distance must be obtained through complex localization techniques, and the use of localization techniques increases the complexity of the agent design. However, for agents navigating in unknown environments, using information about the goal to either form part of the state representation or act as the reward mechanism is usually expensive for both the robot design and for computing costs. This paper proposes using a pre-trained Siamese convolutional neural network (SCNN) to estimate the distance between an agent and its goal, thus enabling agents equipped with onboard cameras to navigate an unknown environment without needing localization sensors. This technique can be applied to environments where a goal location may be unknown, and the only information regarding the goal maybe a description of the goal state. Our experiments show that the Siamese network can learn the distance between the agent and its goal using relatively few training samples. Therefore, it is useful for mapless navigation using only visual state information and reduces the need for complex localization techniques.

**Keywords:** few-shot learning; mapless navigation; reinforcement learning; Siamese convolutional neural networks; mobile robot

## 1. Introduction

Artificial Intelligence seeks to answer the question of how an agent can perceive, understand, predict and manipulate an environment it is placed in [1]. An autonomous or mapless navigation system, which implicitly performs localization and mapping, equips the mobile robot to determine its actual position within the reference frame environment and autonomously move to the desired target position [2]. However, the traditional navigation approach consists of algorithms that include simultaneous localization and mapping (SLAM), path planning and motion control, as highlighted in References [2,3]. Moreover, these methods rely on high-precision global maps or positioning systems, such as GPS, with high limitations in terms of visual capability in unknown environments. Research in the application of mapless navigation to many dynamic real-world scenarios has attracted attention from the research community [4], with more diverse approaches to finding robust alternative solutions being proposed. Machine learning is a sub-field of artificial intelligence (AI) and is the study of theories and algorithms that mimic how the brain learns. Machine learning (ML) allows machines to mimic intelligence without explicit rules defining their behavior [5]. At its core, machine learning is a function approximation. The field of ML is vast and ever growing but can be broadly divided into three sections, namely:

- Supervised learning methods are concerned with inferring a function from input–output pairs. The learning exhibited by these methods is achieved by optimizing a loss function computed from the output produced from the model and the expected output or ground truth.
- Unsupervised learning techniques attempt to learn the hidden structure present in training data without the use of an explicit error/reward signal. These techniques attempt to learn the structure naturally present in the training set and achieve this without an error or reward signal.
- Reinforcement learning (RL) attempts to learn a policy, that maps percepts to actions by interaction with the environment. The RL algorithms are inspired by behavioral psychology and attempt to answer the question of how an agent placed in an environment can learn to behave optimally.

For a machine learning algorithm to generalize beyond the data it was trained on, the number of data samples used for training must be relatively large. Few-shot learning aims at learning models in environments where large datasets are not readily available because of privacy, safety or ethical concerns [6]. Supervised agents receive training information in the form [x, y], where $x$ denotes a vector of features that describes the problem being solved by the agent. The variable $y$ denotes the ground truth for the observed values in $x$. Similarly, the following also applies:

- For an agent tasked with classifying images, the vector $x$ consists of pixel intensities. The ground truth is a discrete value representing the class that the image belongs to.
- An agent classifying bank transactions as fraudulent or not, uses data related to transactions to form $x$. The ground truth $y$ for each $x$ is either a 1, meaning that the transaction data in $x$ represents a fraudulent transaction or a 0, meaning that $x$ represents a valid transaction.
- A robot navigating in an environment makes use of various sensor readings to form $x$. In the case of the robot navigating, defining a ground-truth becomes challenging.

An agents success or failure rate at a given task is highly related to how well the task is described using the feature vector. When the task to be performed by the agent is not governed by a set of rules or in cases when the construction of such rules are infeasible, methods are required that learn a function mapping $x$ to $y$ through interaction with the environment. Reinforcement learning techniques are able to learn mapping functions directly from the environment the agent is placed in by performing actions in the environment and observing their outcomes. An agent placed in an unknown environment is faced with decision making under uncertainty and the sequential nature of states it observes. The uncertainty of moving between states can be modeled using a Markov Chain (MC). A Markov Decision Process (MDP) is a Markov Chain (MC) with external actions ($A$) and an agent that can act in the environment [7]. A Markov Decision Process (MDP) is a controlled Markov chain described by the 5-tuple ($S$, $A$, $T$, $p$, $r$) [8]. Where in this case the parameters

- $S$ denotes the set of states observable in the environment;
- $A$ denotes the set of possible actions that can be performed;
- $T$ is the set of time-steps in which decisions must be made. When a set of goal states exists the process terminates whenever these states are reached;
- $p$ denotes the probability of transitioning from one state to another, and
- $r$ denotes the reward function.

Reinforcement learning methods are applicable when the agent does not know the transition and reward functions in advance [8]. Reinforcement learning differs from the traditional learning paradigms such as, instead of learning mapping functions by observing large samples of pre-labeled examples, the objective is to learn behavioral policies through interaction with the environment [9,10]. The learner is not told which actions to take but must learn which actions yield the most reward by selecting an action and observing the reward [11]. In the reinforcement learning setting, an agent is placed in an environment

where it can perform a set of actions $A = [a_1; a_2; a_3; ...; a_k]$ where $k$ denotes the number of actions possible in the environment. At each timestep $t$ the agent observes the current state of the environment $s_t$, selects an action $a_t$ and receives a reward $r_t$ that indicates the desirability of the action taken [12]. The task of the agent is to perform actions, observe the resulting states and rewards and learn a control policy through trial and error, which maximizes the agent's reward over time [12,13]. Figure 1 illustrates an agent interacting with its environment. The agent observes the state $s_t$ of the environment, performs an action $a_t$, and receives a reward $r_t$.

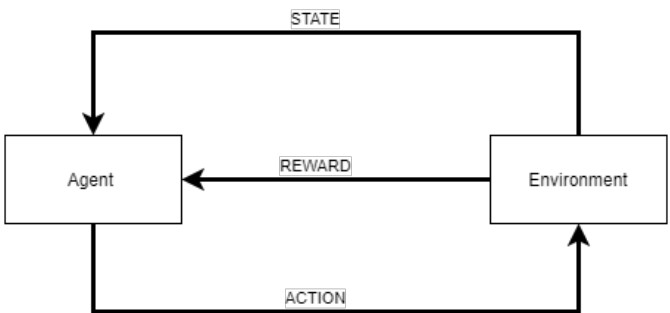

**Figure 1.** An agent interacting with its environment.

The value of starting at an arbitrary initial state $s_t$ and following a policy $\pi$ from that state onward is defined as:

$$V^\pi(s_t) = \sum_{i=0}^{\infty} \gamma^i r_{t+i} \tag{1}$$

A Siamese convolutional neural network (SCNN) is a class of neural network architecture that learns a target function that maps inputs into an output space where a distance metric approximates the semantic distance between the inputs in the embedded feature space [14]. The distance measure may be the Euclidean distance, Manhattan distance, cosine similarity, or Canberra distance [15]. In this study, we make use of the Euclidean distance between the two embedding. Semantic distance is a term often used in natural language processing to refer to the similarity or likeness of two inputs. Given two inputs $x_1$ and $x_2$, we expect that the semantic distance between the two should be small if $x_1$ and $x_2$ are similar and large if otherwise. When using SCNNs, this notion of distance is encoded in the output layer which measures the similarity between the two feature vectors produced from the convolutional layers. Each input $x$ to the Siamese neural network is fed into a separate but identical convolutional network, making the two networks the same ensures that if $x_1$ and $x_2$ are from the same class they cannot be placed far away from one another in the output space [16]. The Siamese CNN is depicted in Figure 2.

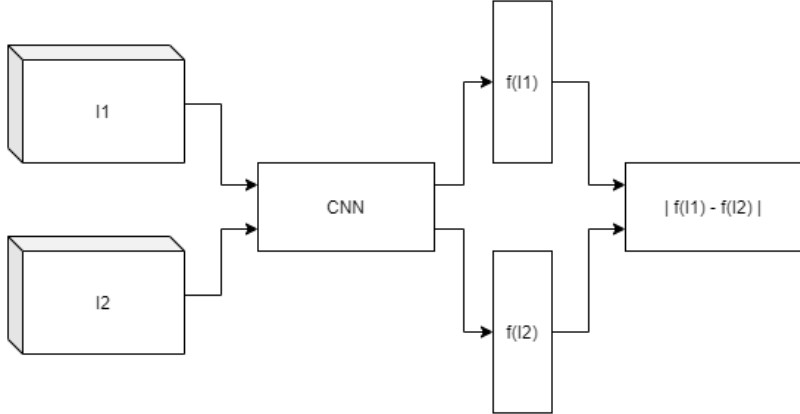

**Figure 2.** Siamese convolutional neural network.

Given two images $I_1$ and $I_2$, and a model $M$, parametrized by a set of weight and biases $\theta$. The SCNN produces the distance between feature vectors according to Equation (2), where $f(I_1)$ and $f(I_2)$ are the outputs produced from the convolutional layers of the SCNN.

$$M(\theta, I_1, I_2) = ||f(I_1) - f(I_2)||_1 \tag{2}$$

where,

- $\theta$ represents the SCNN weights and biases;
- $I_1$ and $I_2$ represent the goal and state images, respectively;
- $f$ denotes the convolutional feature encodings obtained from the SCNN.

Let $Y = 0$ when inputs $I_1$ and $I_2$ represent the same class and $Y = 1$ otherwise. The SCNN attempts to learn a function such that $M(\theta, I_1, I_2)$ is small when $Y = 0$ and large when $Y = 1$.

Navigation in an environment is a fundamental capability of intelligent organisms [7]. This study aims to demonstrate how a mobile agent equipped with an onboard camera and a pre-trained Siamese convolutional neural network, can navigate from an arbitrary start state to a goal state. We assume that the goal/goal state is provided to the agent beforehand. In Equation (1) the sequence of rewards $r$ is generated by starting at a state $s \in S$ and following policy $\pi$ until the episode terminates [13]. This value corresponds to the expected gain according to the specified criterion [8]. In this article we explore the use of a SCNN model to estimate a reward function that is able to guide a mobile robot equipped with an onboard camera from a start state to a goal state. For mapless navigation, a reward function defined by the distance between the agent and its goal is $r_t = \frac{1}{dist(s_t, G)}$. Substituting the new definition of reward into the definition of a states value, we obtain a value estimate that takes into account the distance between the agent and its goal (Equation (3)).

$$V^\pi(s_t) = \sum_{i=0}^{\infty} \gamma^i \frac{1}{dist(s_t, G)}_{t+i} \tag{3}$$

where,

- $\pi$ represents a policy; a function mapping states to actions;
- $V^\pi(s_t)$ denotes the agents value estimate; how much reward can be expected from starting in state $s_t$ and following policy $\pi$;
- $dist(s_t, G)$ represents the Euclidean distance between the current state and the goal state.

Equation (3) provides a value estimate that is large when the distance between the agent and its goal is small, and small as the agent moves further away from its goal; this definition of value assumes that the distance between the agent and its goal $dist(s_t, G)$ can be found at every timestep $t$. In this work, we demonstrate modifying the value estimate by replacing the reward estimate with the similarity estimate produced in Equation (2) to obtain Equation (4). This provides us with a 'distance' estimate that does not rely on localization techniques, and it is accessible at every $t$ and can guide our agent towards its goal $G$.

$$V^\pi(s_t) = \sum_{i=0}^{\infty} \gamma^i \frac{1}{M(\theta, s_t, G)}_{t+i} \tag{4}$$

The main technical contributions of this paper can be summarized as follows:

- Identify a reformulation of an agent's value function that takes into account the distance between the agent and its goal;
- Propose the use of a SCNN to estimate the distance between an agent and its goal;
- We demonstrate that the distance function can used to guide the agent towards its goal;
- We demonstrate that given a relatively small sample size for training, the Siamese convolutional neural network is able to outperform state-of-the-art convolutional neural networks pre-trained on large samples with complex architectures. In this study, we make use of ResNet18 and a KNN baseline model for comparison.

The rest of the article is structured as follows: Section 2 provides an overview of the recent related work from the mapless navigation literature. In Section 3, a discussion of the materials and methods used in this study is presented. A summary of the experimental results and discussion of the study findings is discussed in Section 4. Finally, the concluding remarks and future research directions are presented in Section 5.

## 2. Related work

A fundamental capability of robots that operate in the real world is obstacle avoidance. Obstacle avoidance is typically tackled by approaches based on ranging sensors [17,18]. In Reference [17], the authors proposed a dueling deep double Q-network for obstacle avoidance in indoor environments using only a monocular camera; their approach takes as input an RGB image and outputs linear and angular steering commands. The researchers designed their reward signal to ensure that the robot moves/explores as fast as possible, by penalizing the agent for simply rotating on the spot. This demonstrates how agent behavior can be affected by both the state representation and the reward signal.

In their paper, ref. [19] make use of 174,554 images to train a convolutional neural network to output one of six steering instructions. The researchers report that the network is able to guide the agent, namely a micro aerial vehicle (MAV) from a start state to a goal state 70–80 percent of the time. This demonstrates that supervised methods are able to be applied to mapless navigation tasks at the cost of large amounts of training data needed to fine-tune a pre-trained CNN. Furthermore, the authors demonstrated how the navigation task can be solved using a supervised learning approach.

Departing from traditional supervised learning techniques for navigation, Reference [20] make use of fuzzy logic rules combined with tabular Q learning to solve the task of mapless navigation in static environments. The authors encode domain expertise into the design of their agent in the form of fuzzy logic rules that guide navigation in the environment. Inputs to the model are the distance to the agent's goal and the angle between the agent and goal, the model then outputs steering angle and velocity. The authors in [20] used the number of episodes before the agent finds the goal to measure the performance of their agent.

Learning from exploration eliminates the need for complex pre-trained models that require large amounts of training data. The authors in [21] used a deep Q-network to demonstrate how machine learning can be applied to learn to control a robotic manipulator from visual input only. The DQN requires no prior knowledge aside from the number of actions available to the agent [21]. The authors made use of the distance change between the agent's end-effector and the goal as reward signal at each time step t.

To solve the mapless navigation task, the authors of [22] used sensor fusion, in which laser scan readings are combined with RGB-D images. The inputs to their Asynchronous Advantage Actor-Critic network are the last four sensor values and the orientation to the agent's goal. Training of their agent is done in simulation for 20,000 episodes and the use of a turtlebot to test their algorithm in a real-world setting.

In [23], the authors employed the concepts of a deep Q network to solve the mapless navigation task. The authors pre-train a model consisting of two convolutional layers followed by two fully connected layers in simulation and transfer the model to a real robot. The agent was trained for 50,000 episodes in a simulation using only images as state representation. The researchers verify the feasibility of training deep reinforcement learning agents in simulation and transferring it to real-world robots for autonomous navigation in the environments that it was not originally trained on.

Visual navigation is the problem of navigating in an environment using only camera input. In their paper, the authors of [24] used a modified version of the Actor-Critic algorithm to demonstrate autonomous indoor navigation. Their network was designed to improve the visual navigation task by incorporating image segmentation and depth map prediction. The researchers were able to speed up training by using pre-trained networks to achieve these tasks. At a given time step, the reward received by the agent in [24] is

defined as the pixel difference between the observed states at $s_t$ and $s_{t+1}$. To solve the problem of partial observability, the agent makes use of a long short-term memory (LSTM) cell and demonstrate that this out-performs stacking of recent frames. The authors used 20,000 images from the SUNCG dataset to pre-train their network over the course of two days. Similarly, the authors in [25] proposed the implementation of a few-shot adaptation of visual navigation skills to new observations using meta-learning. Additionally, the authors introduced a learning algorithm that enables rapid adaptation to new sensor configurations or target objects with a few shots. Notably, their results showed that their approach was able to adapt the learning navigation policy with only three shots for unseen situations using different target colors.

In [10], an Actor-Critic network which accepts as input information regarding the current state of the agent and the goal was implemented. The researchers concatenated two convolutional feature vectors using the Hadamard product. The researchers demonstrated using a car navigating in a city as their agent. More so, that combination of state information with goal information allows the agent to learn near-optimal navigation policies. The state representation of the agent at a time step t is defined by a 256 by 651 gray scale image. The autonomous car agent in [10] can perform three actions in its environment, namely, turn left, move forward, and turn right. The authors focused on multigoal learning, and limit their training and testing to the CARLA simulation environment.

To navigate in an environment without a map, Reference [26] make use of an LSTM-based A2C network which accepts as input 30 laser scan readings combined with the distance and angle to the agent's goal. The authors demonstrated that the use of an LSTM network enables state-of-the-art navigation and generalization in the partially observable navigation setting. The authors of [26] make use of 2D simulators to train their model in several different environments before placing their learned network parameters onto a real robot.

To solve the problem of goal-driven mapless navigation in both aerial and water environments, Reference [27] proposed two Actor-Critic-based network architectures. The networks proposed accept as input 20 range readings, linear velocity of the agent, agent's altitude, yaw, the agent's relative position with regards to the target/goal and the agent's relative angle with regards to the goal. The researchers make use of ROS and Gazebo to train their model in two environments. The agent is trained for 1000 episodes and 2500 episodes, respectively. In Reference [28] the authors make use of A Deep Deterministic Policy Gradient network that maps depth images to actions. The authors highlight that to achieve goal-oriented navigation, the state information $s_t$ must be combined with information about the goal.

Reviewing the latest research done on mapless navigation, it becomes apparent that the distance between an agent and its goal plays a very important role in the outcome of the navigation task, this distance is not always known. Supervised learning approaches to navigation as demonstrated by [19] show that using enough data samples, an agent is able to learn the policy mapping visual input to steering commands. The approach taken is 80 percent accurate but requires large amounts of training data. A Deep Neural Network (DNN) is a universal function approximator that can approximate any function given enough data. In their paper, [20] demonstrate how the task can be solved using tabular Q-Learning, which is able to approximate the policy mapping states to actions using a lookup table approach. A state is encountered and the action with the highest $Q$ value is selected and performed. When using tabular Q-Learning, policies that are able to be learnt are not as expressive as using a DQN and it imposes the limitation that the state representation $s_t$ needs to be in a form that can fit into a table. The authors of [21] demonstrate how visual input can be processed using a DQN to move a robotic manipulator. Along with the state representation $s_t$, it becomes apparent that an agent's behavior may also be modified by changing the representation of the reward signal $r_t$. In the mapless navigation setting the reward $r_t$ is usually a function of the agent and its goal [10,20–22,26]. In this article we demonstrate how the need for many data samples demonstrating the mapless navigation

task can be removed and learned by the agent through trial and error. We show that using a relatively small dataset a reward function can be learned that rewards the agent for states close to the goal and penalizes states far from the goal.

## 3. Materials and Methods

Mapless navigation in the domain of robotics refers to navigation in an unknown environment without the use of a predefined model of the environment. Mapless navigation can include object avoidance and a goal state. The former serves to navigate an agent in an environment while avoiding collision with obstacles in the environment. Mapless navigation with the inclusion of a goal state aims to navigate an agent in an environment from an arbitrary start state to the goal state. Traditional supervised learning methods have been applied to solve the problem but require large amounts of training data to generalize unseen environments. Recently, few-shot learning method has been proposed to solve the problem of data deficiency. Few-shot learning allows for function approximation from fewer examples, in this work we propose the use of a Siamese convolutional neural network that makes use of two identical neural networks to learn feature embedding. Reinforcement-learning-based approaches to mapless navigation have relied on the distance to the goal state being known a priori, or that the distance to the goal can be obtained at each timestep. In simulated environments, obtaining the distance to the goal is a trivial task but when applied to the real-world the distance to the goal must be obtained through complex localization techniques. The use of localization techniques increases the complexity of the agent's design. For agents navigating in unknown environments, using information about the goal to either form part of the state representation or act as the reward mechanism is expensive in terms of robot design and computing cost. In this work, we make use of a pre-trained Siamese convolutional neural network to navigate our agent from a start to goal state. We demonstrate that the Siamese network is able to learn the 'distance' between the agent and its goal using relatively few samples for training, allowing for mapless navigation using only visual state information thus reducing the need for complex localization techniques.

### 3.1. Data Collection

For training of the proposed SCNN we use a dataset of 89 images. Our network is trained using goal-background pairs. Examples of samples from the training data are shown in Figures 3–5.

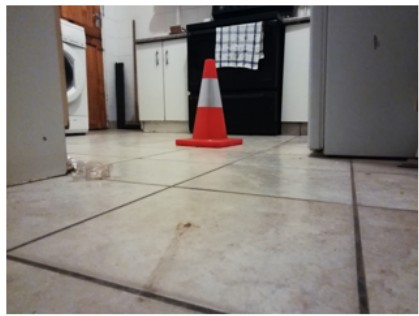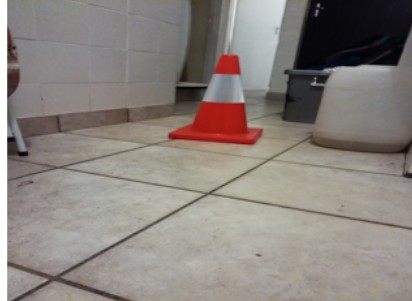

**Figure 3.** Samples labelled 'goal'.

The samples shown in Figure 3 belong to the 'goal' class as a result of having the goal present in the frame.

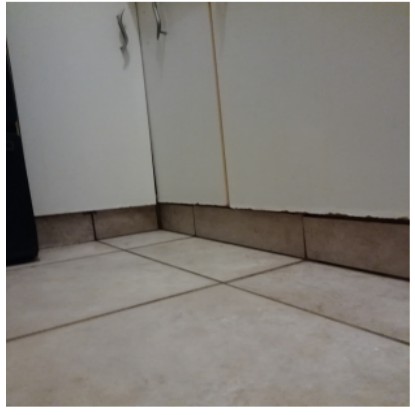 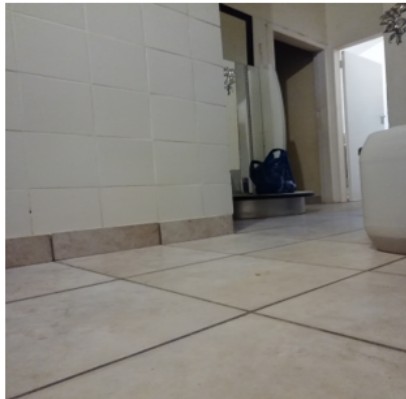

**Figure 4.** Samples labeled 'background'.

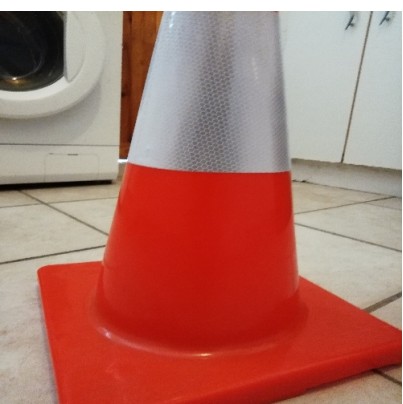

**Figure 5.** The agent's goal state.

We apply up–down flips combined with left–right flips and 45 degree rotations to increase our sample size. Example augmentations are shown in Figure 6.

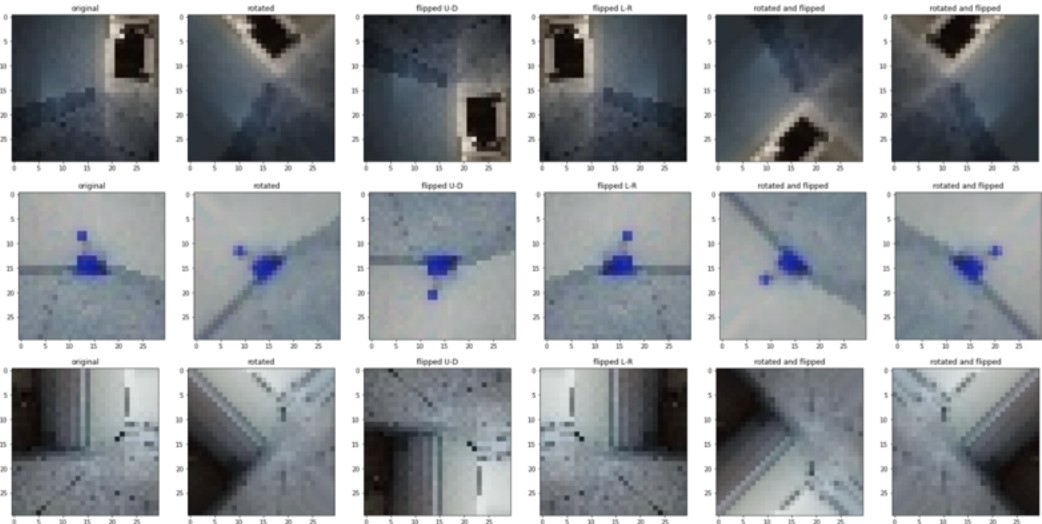

**Figure 6.** Image augmentations for three example images.

### 3.2. Proposed Solution Architecture

The Siamese convolutional network is a convolutional neural network that accepts two inputs and outputs a dissimilarity score for the two inputs. In our case the two inputs during training are the 'background' and 'goal' images. We used the dissimilarity score to replace $dist(s_t, G)$. We set the ground truth for images belonging to the same class

as 0 and 1 for images from different classes. We made use of transfer learning to test a RESNET18-based Siamese network. The output layer is modified to provide a dissimilarity estimate by replacing the final layer with a fully connected layer that outputs a 100-D vector. We also tested our own custom network that consists of four convolutional layers and one max pooling layer. We trained our Siamese networks using a dataset of 89 images using the contrastive loss. For our custom network we used a constant learning rate of 0.0001 and a batch size of 2 for 120 epochs. For the pre-trained networks we made use of a cyclical learning rate. We fine-tuned the ResNet18 model by freezing all the layers and only allowing parameter updates in the final layer, the training and validation curves are shown in Figure 9. Using ResNet18 we also fine-tuned the network end-to-end by allowing parameter updates through the entire network, the results are shown in Figure 10. To provide dissimilarity estimates, the network is modified to output a 100-D feature vector and then calculates the Euclidean distance between the two encodings. We started training with a learning rate of 0.0001 decreasing the learning rate by a factor of 0.1 every 50 episodes. After 100 epochs the loss was oscillating around a single value and we stopped the training.

During training the Siamese network calculates the distance between the convolutional feature encodings produced by each image, namely, the goal and state. This encodes a distance measure into the calculation of the reward. The DQN accepts the current state $s_t$ and provides Q-value estimates for each possible move. The possible moves are forward, rotate left, rotate right. The DQN outputs three values corresponding to each possible action, namely, Q($s_t$, left), Q($s_t$, forward) and Q($s_t$, right). We use an epsilon-greedy action selection strategy with an epsilon of 0.6 during training to allow the agent to search its environment. The epsilon-greedy strategy ensure that the agent does not just select the action corresponding to the maximum Q-value for each state, an epsilon of 0.6 means that the agent will choose a random action with probability 0.6. Our DQN consists of two convolutional layers, each producing 16 and 32 feature maps, respectively.

The physical agent is implemented using a Raspberry Pi 4 micro controller with 2 GB of RAM. The agent is equipped with two DC motors which are controlled L298N motor controller circuit. We make use of two ultrasonic range sensors used to detect when the agent is close to obstacles, the range readings do not form part of the state representation $s_t$. The state $s_t$ is provided by a Microsoft HD Webcam. The physical agent can be seen in Figure 7. All coding necessary for controlling of motors and implementation of deep Q-network and Siamese convolutional neural network is implemented using Python.

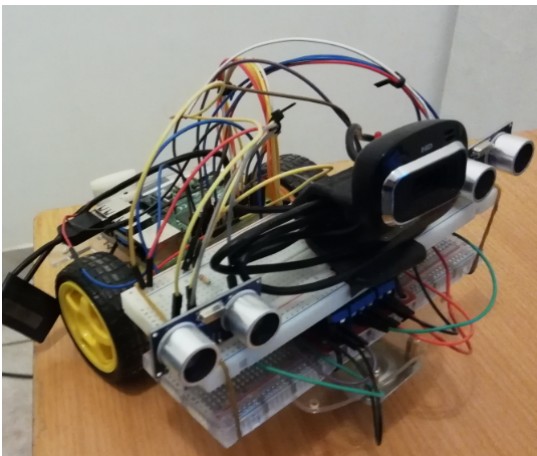

**Figure 7.** Physical agent implementation.

The proposed solution architecture is shown in Figure 8, it consists of a pre-trained SCNN that estimates the reward *r* obtained by our mobile agent. The reward estimates combined with the Q-Value estimates produced by the DQN can be used to guide our mobile agent towards its goal state.

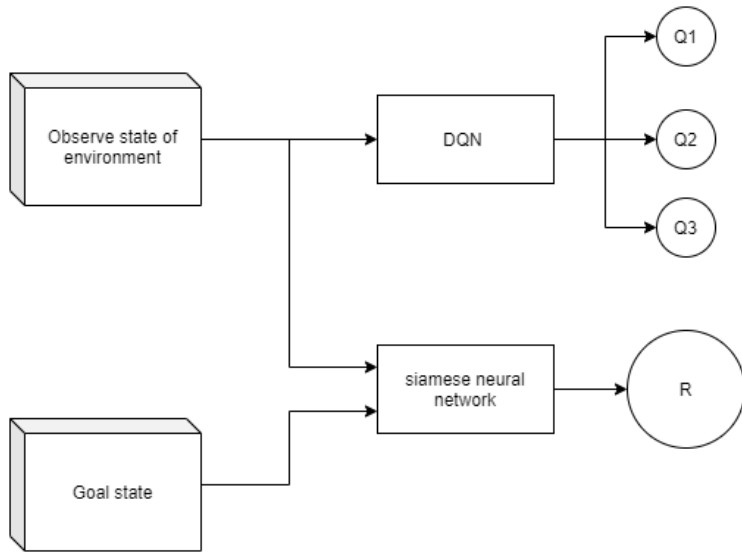

**Figure 8.** Proposed solution architecture.

Our custom SCNN architecture, shown in Figure 2 forms the reward mechanism of our agent. The custom SCNN is used to approximate the distance between the agent and its goal, $dist(s_t, G)$. The DQN network hyper-parameters are summarized in Table 1. The SCNN architecture is summarized in Table 2. The DQN architecture is summarized in Table 3.

**Table 1.** DQN hyperparameters.

| Parameter | Value |
|---|---|
| gamma | 0.999 |
| epsilon | 0.6 |
| alpha | 0.0001 |
| batch size | 4 |
| target network update | 10 |
| max steps per epoch | 200 |

**Table 2.** SCNN architecture.

| Layer | Input Channels | Output Channels | Window/Kernel Size | Padding |
|---|---|---|---|---|
| Conv2D | 3 | 16 | 3 | True |
| Conv2D | 16 | 32 | 3 | True |
| MaxPool2D | 32 | 32 | 2 | False |
| Conv2D | 32 | 16 | 3 | False |
| Conv2D | 16 | 8 | 3 | False |

**Table 3.** DQN architecture.

| Layer | Input Channels | Output Channels | Window/Kernel Size | Padding |
|---|---|---|---|---|
| Conv2D | 3 | 16 | 5 | |
| Conv2D | 16 | 32 | 5 | |
| Linear | 32768 | 3 | n/a | |

## 4. Results and Discussion

In this section, we present the results from training of our proposed solution. Figure 9 shows the training (red) and validation (blue) error over epoch.

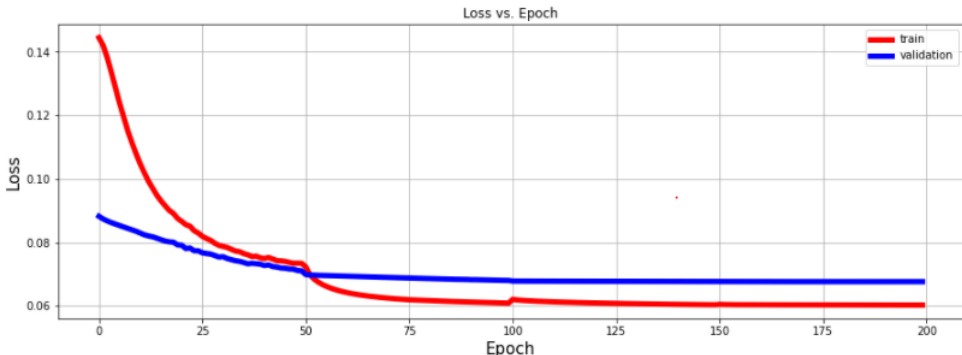

**Figure 9.** Learning curves when finetuning ResNet18 as feature extractor.

The loss over epoch when fine tuning the ResNet18-based network using our dataset is shown in Figure 10. We allowed parameter updates in all the layers of the network, fine-tuning the entire network with our dataset. The learning curves show that the dataset is not sufficiet for training a large network such as ResNet18. We stopped the training loop after 100 epochs.

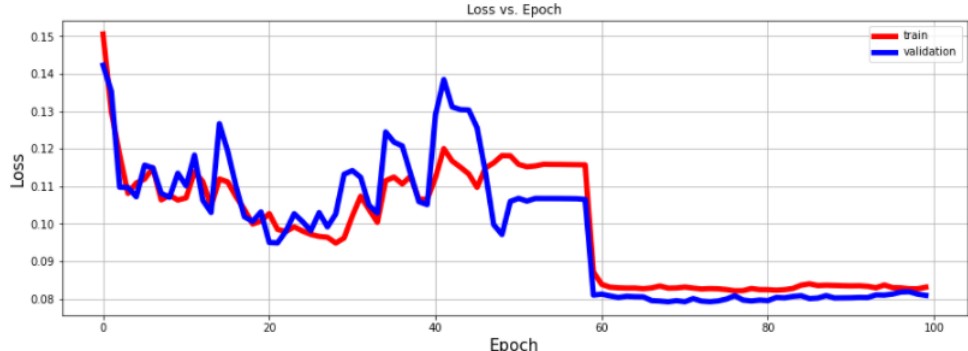

**Figure 10.** Learning curves when training ResNet18-based SCNN.

We trained our own custom network for 120 epochs, the learning curves are shown in Figure 11. After 80 epochs the gap between the learning curves becomes greater, the model was starting to overfit the training data.

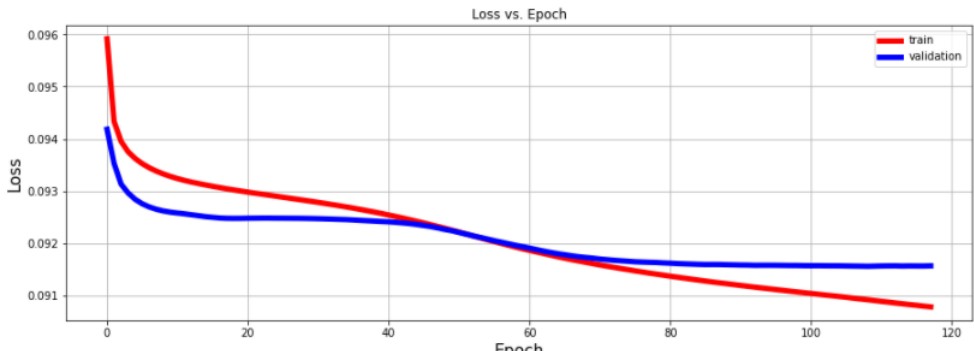

**Figure 11.** Loss over epoch for custom network.

We made use of a threshold value to ensure that states that had the goal present but were far from the goal did not receive any reward. To obtain the optimal value, we tried various values and observed the accuracy score. The results are shown in Table 4. The Siamese network accepts two input images, the network produces a dissimilarity score, the values that fall below the threshold value are predicted as 0 and those that are above the threshold are predicted as 1.

**Table 4.** Accuracy scores for various threshold values.

| Score Threshold | Classification Accuracy |
| --- | --- |
| 0.1 | 49.53 |
| 0.2 | 49.53 |
| 0.3 | 49.53 |
| 0.4 | 49.53 |
| 0.5 | 49.53 |
| 0.6 | 76 |
| 0.7 | 52.8 |
| 0.8 | 50.47 |
| 0.9 | 50.47 |
| 1.0 | 50.47 |

In our case the problem is a regression task and not a classification problem. We make use of classification accuracy and MSE to select a model for final use. The MSE measures how 'tight' predictions are to the ground truth, in our case we require a model that produces scores close to zero for states that are very similar and scores close to 1 for states that are very dissimilar. Results of testing the network when no goal is present in the state are shown in Figure 12. The image on the right is the agent's goal, the image on the left is the current state the agent is perceiving through its camera.

Figure 12 shows that using our dataset and SCNN architecture, our agent is able to learn a mapping that resembles a distance function, that is the agent that receives less reward for a state that is far away and more reward for states that are closer to its goal. A threshold value (Table 4) is used to ensure that the agent does not receive rewards for states where the goal is present in the field of view but far away in terms of distance. Table 5 presents the training time obtained in the cause of the experiment for the different architectures used in this study.

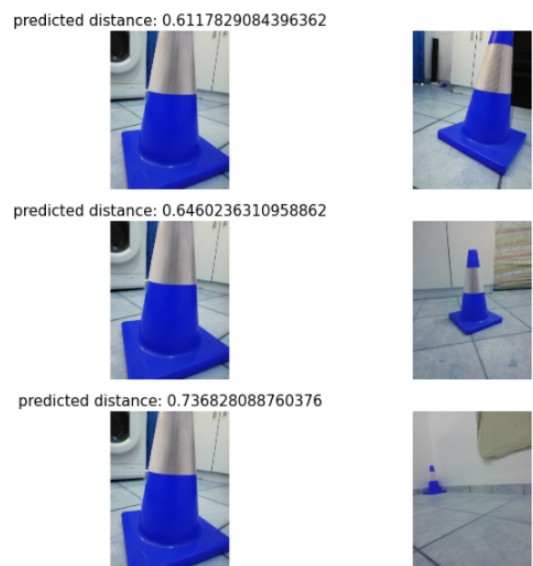

**Figure 12.** Dissimilarity scores for states at various distances from goal.

**Table 5.** Training time for architectures used in this study.

| Algorithm | Average Training Time (s) |
| --- | --- |
| SCNN with ResNet18 base (layers frozen) | 75.046 |
| SCNN with ResNet18 base | 371.412 |
| Custom SCNN architecture | 9.6437278 |

### 4.1. Discussion

We have demonstrated how a Siamese convolutional neural network (SCNN) can be used to estimate the distance between a mobile agent and its goal. The agent is equipped with an onboard camera to capture frames. The results demonstrate that using a relatively small sample of goal–background pairs we can learn a similarity function, this function can then be used to estimate the similarity between an agents current state $s_t$ and its goal. Figures 9 and 10 show the learning curves when using pre-trained models to estimate rewards, Figure 11 demonstrates that with the given data samples a network with fewer parameters is able to learn the function well. Figure 12 shows reward estimates when the agent is provided with its goal state and must estimate reward for states at varying distances from the goal; demonstrating that the learned reward function is able to encode a notion of distance. Table 4 displays classification accuracy scores for various threshold values from 0.1 to 1.0. The inclusion of a threshold value allows the agent to control how similar states must be to the goal to receive reward. States far away from the goal receive less reward than states closer to the goal. We use a Raspberry Pi 4 to implement a physical agent and show that using the SCNN architecture the agent is able to navigate towards its goal.

In this article we have demonstrated that the reward mechanism $r_t$ can be replaced by $\frac{1}{dist(s_t, G)}$ and that the distance between the agent and its goal ($dist(s_t, G)$) can be estimated using a neural network architecture that is able to learn from fewer samples. In our work, we have used a modified form of the value of a state $s_t$ following a policy $\pi$ (Equation (1)) to guide our agent towards its goal. We have shown that the use of a threshold value can be used to control how similar a state must be to its goal in order to receive reward.

### 4.2. Limitations of the Study

Our visual approach to navigation is able to guide an agent from a start state to a goal state using only visual input obtained from an onboard camera. The performance of a visual approach may be affected by obstacles that may be present in the environment. To detect obstacles in the agent's environment, we have made use of ultrasonic range finders to identify when our agent is close to obstacles. As a result of training the agent in the real-world setting, we use a threshold value for the distance readings to avoid physically damaging our agent when it collides with obstacles. We attempted to demonstrate a simpler approach to mapless navigation using only visual sensory input, we have kept our agent design simple and relatively inexpensive. We experienced some difficulty with the longevity of the motors we decided to use to control our agent, this may be handled by training in simulation and transferring learned knowledge to our physical agent, however, a trade-off exists because transferring knowledge from simulation to the real world is challenging. Our agent may receive reward for states in which the goal is present, we combat this by applying a threshold to the similarity score produced by our SCNN, which is used ensure the agent is rewarded for states that are close. Consider cases in which the agent may be very close to the goal but it may be facing in an opposite direction and never obtain a frame where the goal is present. Our agent has been trained for indoor navigation and due to resource constraints we have not investigated outdoor environments and tested whether our agent is able to navigate in unseen environments. This may be alleviated by increasing the sample used for training and providing a more diverse set of examples from varying environment, both indoor and outdoor.

## 5. Conclusions and Future Work

Mapless navigation refers to navigation of an agent in an environment without a model of the environment. The distance to the goal in a mapless navigation task is used to inform the agent's navigation. The agent can use this distance measure to select actions that bring it closer towards its goal. In simulated environments, obtaining this distance is trivial. In real-world settings this distance is found using localization techniques such as

LIDAR and GPS. For agents in the real world not equipped with localization sensors, the mapless navigation task cannot rely on this distance measure to select actions.

In this paper, we have demonstrated how a pre-trained SCNN can be used to estimate the distance between an agent and its goal, enabling agents equipped with onboard cameras to navigate in an unknown environment without the need for localization sensors. This technique can be applied to environments where the location of a goal may be anonymous, and the only information regarding the goal may be a description of the goal state. Examples of such environments are:

- An agent tasked with locating a bomb may not know where the bomb is but may have some examples of what a bomb looks like;
- An agent tasked with helping an elderly person navigate their environment may not know where all the obstacles/items are located around it but it has a description given to it of what it needs to find;
- An agent tasked with guarding a premises from intruders may not know where the intruder is located but given examples of previous intrusions, it can locate and detain the intruder.

As future research perspectives, it is worth noting that our agent was able to make informed decisions that bring it closer to its goal using the SCNN. Therefore, we would like to highlight here some promising areas of future improvement upon our agent design, namely:

- LSTM layers in the DQN architecture would enable a sort of memory for an agent and would make more of the environment 'visible' to the agent at each $t$. The inclusion of LSTM layers would increase the complexity of the network and may increase time needed to learn the task.
- Using visual input as a state representation can be difficult when the goal is blocked by obstacles in the environment and is not visible to the agent. In future works we would like to incorporate more information into $s_t$ to overcome the problem of partial observable environments.
- In this article the two networks used are treated separately. In future work, the networks may be combined and their parameters updated as one. This would enable learning the task 'on-the-fly'.
- In this work, we have made use of a pre-trained SCNN which may still require effort to collect a few samples of data for pre-training. We have taken a few-shot learning approach, in future works one-shot and zero-shot approaches can be explored to reduce the need for a dataset for pre-training.
- We make use of the Euclidean distance to calculate the distance between the embedding produced by the SCNN. In future we would like to explore the effects of various distance metrics on the navigation task.
- Imitation learning is a technique to learn models from a human teacher. In future works on mapless navigation we would like to explore a few-shot variant of imitation learning which is able to learn navigation policies from a single demonstration of navigation in an environment.
- The design of our agent was kept simple to demonstrate cost-effective mapless navigation in indoor environments. This may be enhanced to include outdoor environments with varying terrain and conditions by incorporating larger wheels with a track or tyre and stronger motors.
- The current study focused primarily on the DQN algorithm as a result of its relatively simple and intuitive nature. Since its inception, the DQN has been improved upon and as such we would like to test how our method performs when used with other algorithms from the RL literature.
- Multi-agent systems make use of multiple agents working together to achieve a goal. Multiple agents contributing towards the same goal may be useful in environments that are large or complex to navigate or where there are multiple goals to achieve. In future works we would like to incorporate mobile robots with drones. The mobile

robot would observe the states obtainable from the ground and the drones obtained from the air. The drone would also allow for more of the state to be visible to the ground agent at the same time.

**Author Contributions:** Conceptualization, V.K., M.O. and A.E.; methodology, V.K.; software, V.K.; validation, V.K., M.O. and A.E.; investigation, V.K.; resources, V.K.; data curation, V.K.; writing—original draft preparation, V.K., M.O. and A.E.; writing—review and editing, V.K., M.O. and A.E.; supervision, M.O. and A.E. All authors have read and agreed to the published version of the manuscript.

**Funding:** This research received no external funding.

**Institutional Review Board Statement:** Not applicable.

**Informed Consent Statement:** Not applicable.

**Data Availability Statement:** All data used for the study are mentioned within the manuscript. Moreover, the study did not report any new data.

**Acknowledgments:** Not applicable.

**Conflicts of Interest:** The authors declare no conflict of interest.

## Abbreviations

The following abbreviations are used in this manuscript:

| | |
|---|---|
| RL | Reinforcement Learning |
| SCNN | Siamese Convolutional Neural Network |
| Conv2D | 2-D Convolution |
| MaxPool2D | 2-D Max Pooling |
| MSE | Mean Squared Error |
| DQN | Deep Q-Network |
| DNN | Deep Neural Network |

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
