# Peer review of "A Few-Shot Learning-Based Reward Estimation for Mapless Navigation of Mobile Robots Using a Siamese Convolutional Neural Network"

_applsci, doi:10.3390/app12115323_

Round 1
Reviewer 1 Report
Both few shot learning and navigation are important issues in the research community. this paper proposes a mapless navigation that can navigate an agent in an environment while avoiding collision with obstacles in the environment. The pre-trained Siamese CNN is used to guide the agent moving from the start to the goal state, and results demonstrate its effectiveness and efficiency. Overall, the paper contains novelty and make contributions. Here are some points that may help improve the qualiity of the manuscript: 1. it is better to make necessary comparisons to well-known baselines; 2. some important references should be included to make it more complete, e.g., enabling smart urban service with gps trajectory data; 2TD Path-Planner: Towards a More Realistic Path Planning System over Two-Fold Time-Dependent Road Networks; CrowdExpress: A Probabilistic Framework for On-Time Crowdsourced Package Deliveries;
Reviewer 2 Report
In this work the authors make use of a pre-trained Siamese convolutional neural network to navigate our agent from a start to goal state.
The motivation is the following
Reinforcement learning techniques have relied on the distance to the goal state being known a priori,
or that the distance to the goal can be obtained at each timestep. In simulated environments, obtaining
the distance to the goal is a trivial task but when applied to the real-world, the distance to the
goal must be obtained through complex localization techniques. The use of localization techniques
increases the complexity of the agent’s design. For agents navigating in unknown environments,
using information about the goal to either form part of the state representation or to act as the reward
mechanism is expensive in terms of robot design and computing cost.
Results
The authors demonstrate that the Siamese network is able to learn the ‘distance’ between the agent and its goal, allowing for mapless navigation using only visual state information thus reducing the need for
complex localization techniques.
The background of the work is clear, quoting "Mapless navigation in the domain of robotics refers to navigation in an unknown environment without the use of a predefined model of the environment. Mapless navigation can include object avoidance and a goal state. The former serves to navigate an agent in an environment while avoiding collision with obstacles in the environment. Mapless navigation with the inclusion of a goal state aims to navigate an agent in an environment from an arbitrary start state to the goal state." In this scenario, the authors give a relevant contribution since they provide some theoretical contribution and also real experiments. At the same, the authors may consider related work: Towards tactical behaviour planning under uncertainties for automated vehicles in urban scenarios; Towards tactical lane change behavior planning for automated vehicles. The relevance of this is to consider MDP (as the authors do), and use MDP for navigation.
Other comments follow
please clarify why the L1 norm is used in Siamese CNN instead of L2 norm or other norm
Is "adaption" correct? maybe adaptation? or adoption?
I suggest the authors to explain the meaning of "Few-Shot Learning" in the abstract
Maybe the title should mention the Siamese network?
Is the caption of Fig. 10 complete? "Learning curves when finetuning ResNet18"
If possible, provide the training time. As far as I know, the training time of CNN can be very long
The references are not complete in the sense that journal names are missing (pages, volume, etc). In addition, the authors may want to consider more references
Reviewer 3 Report
Please, refer to the attached document!

Reviewer 4 Report
Reviewer's summary after reading the manuscript:
When applied to robotics, mapless navigation is described as navigating in an unfamiliar environment without the use of a previously specified model of the environment. Mapless navigation may entail object avoidance as well as the achievement of a desired state. When an agent is in an environment, the former helps to direct the agent through the environment while avoiding collisions with barriers in the environment. Mapless navigation with a target state is a navigation technique that seeks to guide an agent across an environment from an arbitrary start state to a goal state. Reinforcement learning strategies have traditionally depended on either the distance to the target state being known a priori or the ability to acquire the distance to the goal at each timestep. While determining the distance to the objective in simulated settings is a simple operation, when applied to the actual world, the distance to the goal must be achieved using complicated localization procedures. Using localization methods to construct an agent makes it more difficult to create a simple design. The use of goal information for agents navigating in unfamiliar settings, either as part of the state representation or as a reward mechanism, is both time-consuming and costly in terms of robot design and processing resources. In this study, we make use of a pre-trained Siamese convolutional neural network to guide our agent from a start state to a target state. The authors show that the Siamese network is capable of learning the 'distance' between the agent and its objective, allowing for mapless navigation using just visual state information, hence avoiding the requirement for sophisticated localization algorithms.
----------------------------------------
Dear authors, thank you for your manuscript. I enjoyed reading it. Presented are some suggestions to improve it:
(1) Please consider modifying the title of the manuscript to include the words "robotics" and " Siamese Convolutional Neural Networks" so that it would be easier for potential readers to find your study.
(2) Please include a "Limitations of the study" section to discuss what were the challenges faced, and how your team overcame those challenges. This would be very beneficial to the readers as they would be able to learn from your expert knowledge.
(3) To improve the impact and readership of your manuscript, the authors need to clearly articulate in the Abstract and in the Introduction sections about the uniqueness or novelty of this article, and why or how it is different from other similar articles.
(4) The Conclusion section is too short. Please expand it by discussing the future directions of your research, especially how it may contribute to your ongoing research.
(5) Please substantially expand your review work, and cite more of the journal papers published by MDPI.
(6) Some of the references cited are not yet properly formatted. For example, the DOIs of all the journal papers cited are not included yet. For the references, instead of formatting "by-hand", please kindly consider using the free Zotero software (https://www.zotero.org/), and select "Multidisciplinary Digital Publishing Institute" as the citation format, since there are currently 21 citations in your manuscript, and there may probably be more once you have revised the manuscript.
Thank you.
Round 2
Reviewer 1 Report
I have no further comments.
Reviewer 3 Report
please refer to the attached document!
